# Shape registration in the time of transformers

**Giovanni Trappolini**
Department of Computer Engineering
Sapienza University of Rome
giovanni.trappolini@uniroma1.it

**Luca Cosmo**
DAIS
Ca' Foscari University of Venice
luca.cosmo@unive.it

**Luca Moschella**
Department of Computer Science
Sapienza University of Rome
luca.moschella@uniroma1.it

**Riccardo Marin**
Department of Computer Science
Sapienza University of Rome
marin@di.uniroma1.it

**Simone Melzi**
Department of Computer Science
Sapienza University of Rome
simone.melzi@uniroma1.it

**Emanuele Rodolà**
Department of Computer Science
Sapienza University of Rome
emanuele.rodola@uniroma1.it

## Abstract

In this paper, we propose a transformer-based procedure for the efficient registration of non-rigid 3D point clouds. The proposed approach is data-driven and adopts for the first time the transformer architecture in the registration task. Our method is general and applies to different settings. Given a fixed template with some desired properties (e.g. skinning weights or other animation cues), we can register raw acquired data to it, thereby transferring all the template properties to the input geometry. Alternatively, given a pair of shapes, our method can register the first onto the second (or vice-versa), obtaining a high-quality dense correspondence between the two. In both contexts, the quality of our results enables us to target real applications such as texture transfer and shape interpolation. Furthermore, we also show that including an estimation of the underlying density of the surface eases the learning process. By exploiting the potential of this architecture, we can train our model requiring only a sparse set of ground truth correspondences ($10 \sim 20\%$ of the total points). The proposed model and the analysis that we perform pave the way for future exploration of transformer-based architectures for registration and matching applications. Qualitative and quantitative evaluations demonstrate that our pipeline outperforms state-of-the-art methods for deformable and unordered 3D data registration on different datasets and scenarios.

## 1 Introduction

Recent technological advancements of 3D acquisition pipelines have produced an abundance of available data. The direct consequence is the non-standardization of the acquisition process. Such technological democratization brings along a disparate amount of different representations, discretization, and arbitrary resolution. Given so, the request to align such data has become urgent. Furthermore, data-driven statistical approaches require aligned data to relate feature changes across the population, inferring underlying patterns.

The Computer Vision community has devoted an extraordinary effort in the last decades to address 3D objects analysis. A common way to approach this problem is to align the geometry of one

35th Conference on Neural Information Processing Systems (NeurIPS 2021).

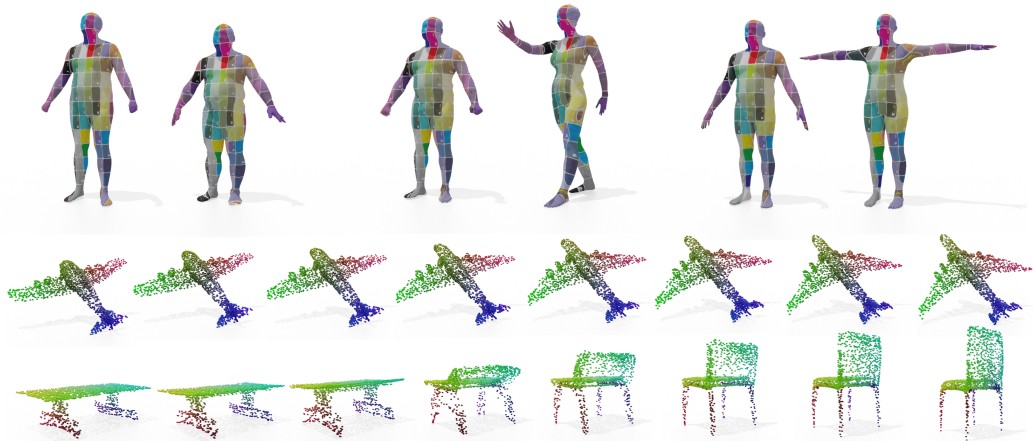

Figure 1: Some results in real application targeted by our method. In the first row, three examples of texture transfer on human shape pairs. In the second and third row, two examples of interpolation between intra-class and inter-class shapes from ShapeNet (one for each row).

known shape to an incoming one. Such methodology is referred to as *registration*. Many different axiomatic pipelines have been proposed that address different kind of objects and domains. While many methods rely on the assumption that the shapes used in the registration task differ just by a rigid transformation, the non-rigid domain is far more complex and interesting. This category pertains organic objects (e.g., humans, animals, internal organs), which are particularly interesting as well. Non-rigid registration aims to align two geometries that may differ by bending and stretching of the geometry, which may also significantly modify its metric. This problem is even more complicated if the geometry representation is given just by a sparse point cloud.

However, an emerging field merges data with classical algorithmic problems, exploiting such statistics as regularization. Among the different learning approaches, recently, the use of the Attention mechanism has become significantly popular in NLP domain, being later transferred to Computer Vision applications. Such architecture is called Transformers, and they represent one of the most significant groundbreaking methodological advancement since the introduction of CNNs.

In this work, we aim to let non-rigid registration meet the transformers. Intuitively, we aim to use the transformer as a *geometrical translator* between two non-rigid point clouds. As the first element, we modified the attention mechanism, proposing to make it aware of the underlying density of the geometry. Hence, we apply such a mechanism in an autoencoder–like architecture, which takes a template point cloud as input and aims to modify its geometry to fit the target point cloud.

The proposed method achieves better results than several state-of-the-art competitors in the shape matching task. We show results on humans case, but also inter-class objects. In this second case, our method is trained in an unsupervised manner, showing the power of our attention mechanism to infer the underlying geometry. Also, thanks to the attention mechanism, we are able to interpret what the network considers relevant for the registration. Finally, we can target texture transfer and shape interpolation showing applicability in real tasks as demonstrated in 1.

Our contributions could be summarized as follows: **(a)** We propose the first transformer for non-rigid registration task, showing the advantages of translation paradigm; **(b)** We modified the attention mechanism to make it aware of the point cloud density and of the underlying geometry of a shape; **(c)** We significantly improve the state-of-the-art performances on different datasets and challenging scenarios.

All code and data is publicly available [1].

---

## 2 Related work

Shape matching is a problem with a tradition of decades. For a complete overview, we refer to the surveys [54, 51]; below we cover the literature that more closely relates to our work.

**Surface matching** Early attempts to match non-rigid objects work under the assumption of near isometry. Such is the case, for instance, of blended intrinsic maps [27], which combine multiple conformal mappings with an additional penalty to preserve local areas. Similarly, several variants and applications of the functional maps framework [41] implicitly assume near isometries by requiring a special structure of the functional representation, or by means of dedicated regularizers [42, 40, 18, 38, 49] also designed for handling partial shapes [50, 11, 45]. A common drawback to these works is that they do not disambiguate the intrinsic shape symmetries; further, the surface connectivity introduces a structural bias which may affect the performance, as recently shown in [36]. An attempt to overcome these issues was proposed in [37], but with the extra assumption that the shapes to match are in the same pose. More recently, SmoothShells [15] proposes an iterative algorithm to recover dense correspondences by an alignment of intrinsic information. While these methods do not address 3D shape registration directly, they rely on the general idea that a correspondence can be recovered by aligning specialized, possibly high-dimensional embeddings of the shapes at hand.

**Template-free registration** A popular approach to solve for a matching between 3D objects is to align their geometries extrinsically via ICP-like procedures [5, 29, 2]. In fact, registration and matching are intimately related problems with different goals. While the matching problem aims to find a *combinatorial* solution, which indicates for each point its image on the target shape, registration looks for a *spatial* transformation of the geometry. If the two shapes have a significantly different discretization, the latter problem is less ambiguous than the former. ICP-based approaches iteratively solve the two subproblems in an alternating fashion, by finding a point-to-point correspondence and the best transformation that adheres to such correspondence. These methods do not convergence to a good solution if the input shapes are significantly misaligned. Similarly, Coherent Point Drift [39] and variants [26, 22] rephrase the registration problem as an alignment of probability densities.

**Template-based registration**

A different family of approaches make use of a given template, which is known a priori and is possibly parametric, toward which a given input shape is to be matched. Taking as an example the case of human bodies, it is common to model the surface deformation related to the subject identity using PCA [1, 3, 31, 44], while recent advancements in statistical data-science suggested that non-linear methods are more expressive to catch fine details of humans [48, 59, 9]. Similarly, the pose can be modeled by simple paradigms like Linear-Blend Skinning [31, 44], triangle deformations [3, 23], but also learning methods [59]. Efforts to register such templates to arbitrary target models have been carried out extensively by the community [23, 61, 32, 33]. However, the requirement of a template is not always easy to satisfy.

**Learning methods** With the rise of learning methods, several attempts have been made to introduce a statistical prior to the matching and registration process. For example, several extensions have been proposed to bring the functional maps formalism into a learning paradigm [30, 13, 53]. SmoothShells has also been extended to be data-driven [16]. The point cloud representation has received comparably less attention, mainly in a rigid alignment setting [24, 43, 52]. In the non-rigid domain, a seminal work is 3DCoded [19] that proposes a proper registration using an autoencoder architecture. However, having a fixed template forbids inter–class operations and limits the use of the point cloud structure. The constraint of having a fixed template has been relaxed in [21] but it still requires couple of isomorphic shapes in training. Recently, it has been proposed to learn a linearly-invariant embedding [34], but the method requires training two separate networks and relies on simple PointNets [46] which are not able to catch the fine details of the objects. Finally, a recent trend in geometric deep learning suggests that implicit representations may also be used for shape matching [6].

**Transformer-based architectures** Transformers have been first introduced in the context of neural machine translation by the pioneering work [56], and later procedeed to revolutionize the field of natural language processing [12, 47]. The success obtained in NLP inspired further work to employ transformers in computer vision [14], where they managed to outperform convolutional networks. Exploiting the input invariance property characterising the attention mechanism, many works have

naturally extended transformers to handle point clouds [20, 60, 17]. These works show promising results, but work only in the context of object classification and segmentation. Instead, [57] proposes a network to find the rigid alignment between two point clouds imitating ICP, and using a transformer architecture to infer the residual term. Differently, we aim to solve for *non*-rigid registration, which is a more general case.

# 3 Notation and general objective

**3D shapes** We model 3D shapes as compact 2-dimensional Riemannian manifolds $\mathcal{M}$, possibly with boundary $\partial \mathcal{M}$. In the discrete setting, we represent the manifold $\mathcal{M}$ as an unorganized point cloud of $n_\mathcal{M}$ vertices embedded in $\mathbb{R}^3$, and encoded in a coordinate matrix $X_\mathcal{M} \in \mathbb{R}^{n_\mathcal{M} \times 3}$.

**Shape registration** The main objective of this paper is to introduce a data-driven approach to perform shape registration. Given a *source* shape $\mathcal{S}$ and a *target* shape $\mathcal{T}$, respectively represented by the sets of vertices $X_\mathcal{S}$ and $X_\mathcal{T}$, our goal is to find a corresponding 3D position for each point in $\mathcal{S}$ on the surface of $\mathcal{T}$. An example is shown in the inset figure, where the underlying surface is visualized for reference only.

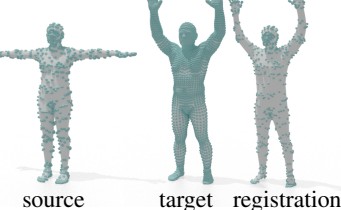

source    target   registration

Our method does *not* assume the source $\mathcal{S}$ to be a fixed template, but can generalize to arbitrary shape pairs $\mathcal{S}$ and $\mathcal{T}$; this is in contrast, e.g., with [19], where the objective is to learn how to deform a fixed known template into another shape.

**Attention** One of the most influential ideas in the recent years, attention originated in the realm of natural language processing but has since gained traction in other fields, such as computer vision and signal processing, due to the vast increase in performance and interpretability exhibited in several tasks. At its heart, the attention mechanism allows learning models to encode latent relations between inputs, assigning higher importance, or "attention", to the parts deemed more relevant. In this sense, attention allows to efficiently capture context information as well as higher order dependencies.

Formally, given two generic input sequences $X_1 \in \mathbb{R}^{n \times d}$ and $X_2 \in \mathbb{R}^{m \times d}$ (for clarity of exposition we assume a constant embedding dimension $d$, but it is not a necessary assumption), the attention mechanism models the linearly encoded representation of the inputs as triplets of query, key, and value matrices, respectively $Q \in \mathbb{R}^{n \times d}$, $K \in \mathbb{R}^{m \times d}$, and $V \in \mathbb{R}^{m \times d}$. The attention score $W$ is then defined as $W = \text{softmax}\left(QK^T d^{-0.5}\right)$ and used to compute the weighted mean of the value vectors, resulting in an output feature matrix $A = WV$. We refer to *self*–attention whenever $X_1$ and $X_2$ are the same object, and use the term *cross*–attention otherwise.

# 4 Method

Our method takes as input two point clouds $X_\mathcal{T}$ and $X_\mathcal{S}$, with $n_\mathcal{T}$ and $n_\mathcal{S}$ points respectively, and deforms the points of the source shape $X_\mathcal{S}$ to fit the geometry described by the target point cloud $X_\mathcal{T}$. To do so, we rely on a novel attention mechanism which considers the underlying geometry conveyed by the point cloud, rather than treating the points simply as elements of a set.

**Surface Attention** The classic attention definition, as introduced in Section 3, looks at the input data points simply as elements of a set, and uses the computed attention scores to perform a weighted sum of the value vectors. When the value vectors represent a sampling of a signal over a surface, however, the natural domain for the integration should be the surface itself, of which the weighted sum is just an approximation highly sensitive to the specific surface sampling (depending, for instance, on the acquisition method).

To overcome this limitation we propose to modify the attention mechanism to consider the portion of surface represented by each point, weighting the attention score by an estimated local area element.

In practice, for each point $x_i \in X$, we estimate its area contribution as the inverse of the local point density: $\mathcal{A}(X)_i = (|\{x_j \in X \text{ s.t. } \|x_j - x_i\|_2 < r\}|)^{-1}$, where $|\cdot|$ denotes the cardinality of a set, and $r$ is a local radius ($r = 0.05$ in our experiments). Note that, we are not interested in the absolute value of the area elements, rather on the relative contribution of each point. The surface attention

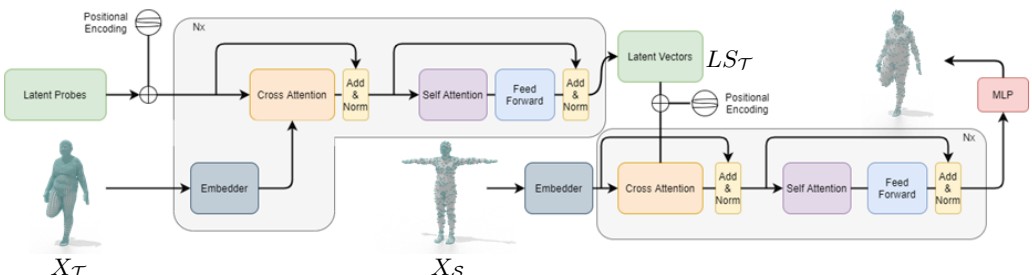

$X_\mathcal{T}$            $X_\mathcal{S}$

Figure 2: The proposed transformer-based architecture for 3D point cloud registration. The latent probes capture the geometry of the input target shape $X_\mathcal{T}$ through the encoder layers. The resulting latent vectors drive the deformation of an input source shape $X_\mathcal{S}$ in the decoding layers, resulting in a deformation of the points of $X_\mathcal{S}$ to fit the geometry of $X_\mathcal{T}$.

score is thus defined as:

$$\widetilde{W}_{i,j} = \frac{e^{W_{i,j}} \mathcal{A}(X)_j}{\sum_t e^{W_{i,t}} \mathcal{A}(X)_t} . \tag{1}$$

As in the classical attention, output features are computed as $\widetilde{W}V$. Figure 3 shows that the surface attention mechanism results in a more stable localization of the attention scores across different samplings of the surface.

In our architecture, each time attention is computed over point clouds, we use such formulation.

**Architecture** The architecture we propose, portrayed in Figure 2, is an *iteratively conditioned* autoencoder. The two main ingredients are an encoder, that maps a target point cloud $X_\mathcal{T}$ into a latent space $LS_\mathcal{T}$, and a decoder, that deforms a source point cloud $X_\mathcal{S}$ to resemble the geometry of the target point cloud $X_\mathcal{T}$.

The encoder draws inspiration from [25], featuring a set of learnable parameters, we refer to them as latent probes $LP$. It presents multiple layers of cross attention which iteratively condition the latent probes $LP$ with the embedding of $X_\mathcal{T}$. After each conditioning, the resulting latent space is further transformed by layers of self attention, feed forwards and residual connections. The encoder output is a set of latent vectors $LS_T$ containing relevant information collected from the target point cloud $X_\mathcal{T}$.

The decoder is analogous to the encoder but the relationship between the latent space and the point cloud in the cross attention is reversed. That is, the embedding of the source point cloud $X_\mathcal{S}$ is transformed and iteratively conditioned with the latent space $LS_T$ produced by the encoder. This procedure induces a deformation of $X_\mathcal{S}$ that, after a final MLP layer, aligns to the points of $X_\mathcal{T}$.

**Training** Given a training dataset equipped with a ground-truth correspondence, we train our network in a supervised setting. Starting from a set of shapes in correspondence, we minimize the reconstruction error using the standard reconstruction loss:

$$L^{sup}(X_\mathcal{S}, X_\mathcal{T}) = \|X_\mathcal{T} - D(X_\mathcal{S}, E(X_\mathcal{T}))\|_2^2 . \tag{2}$$

Furthermore, in case a ground-truth correspondence is not available, our network can be trained in an unsupervised way using the Chamfer distance, between $X_\mathcal{T}$ and $Y_\mathcal{T} = D(X_\mathcal{S}, E(X_\mathcal{T}))$, as defined below:

$$\sum_{s \in X_\mathcal{T}} \min_{t \in Y_\mathcal{T}} \|s - t\|_2^2 + \sum_{t \in Y_\mathcal{T}} \min_{s \in X_\mathcal{T}} \|t - s\|_2^2 . \tag{3}$$

**Testing** At test time, our network can register a point cloud to another instantaneously, with a single forward pass. In the following experiments that involve the computation of matching, the correspondence can be obtained just by looking for the Euclidean nearest-neighbor between $X_\mathcal{T}$ and the output $D(X_\mathcal{S}, E(X_\mathcal{T}))$ in the 3D space.

**Refinement** The peculiar structure of our architecture allows us to refine the output during the testing procedure. This is achieved by minimizing the energy function chamfer$(X_\mathcal{T}, D(X_\mathcal{S}, E(X_\mathcal{T})))$ with

Table 1: Ablation study. We report the relative average matching error (lower is better) with respect to the baseline architecture used in all the experiments (for which the error is set to 1).

| Test Set | # of latent space vectors | | | | | # E/D layers | | | latent space dimension | | |
|---|---|---|---|---|---|---|---|---|---|---|---|
| | 32 | 16 | 8 | 4 | 2 | 4 | 8 | 12 | 32 | 64 | 128 |
| SURREAL | 1.00 | 1.00 | 0.93 | 0.97 | 0.91 | 1.04 | 1.00 | 0.93 | 1.36 | 1.00 | 1.45 |
| FAUST | 1.00 | 1.01 | 1.24 | 1.30 | 1.29 | 1.45 | 1.00 | 1.22 | 0.82 | 1.00 | 0.94 |
| Average | 1.00 | 1.01 | 1.08 | 1.13 | 1.10 | 1.24 | 1.00 | 1.07 | 1.09 | 1.00 | 1.19 |

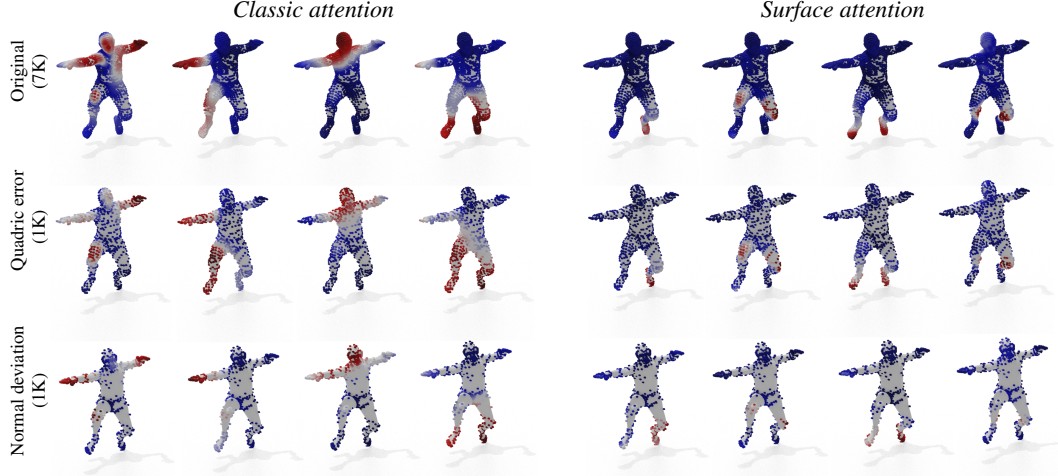

Figure 3: Comparison between the classic (left) and surface (right) attention mechanism behavior with differently sampled surfaces. The surface attention is more stable across different sampling strategies. The two attention mechanism have been trained separately, so there is no correspondence on the attention localization between the two. Surfaces are shown just for visualization purposes.

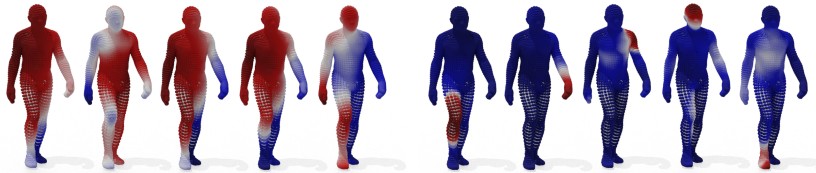

Figure 4: The colormap shows the attention value given to each point of the shape by each input latent vector (only 5 of 32 are shown) at the first (left half) and last (right half) layers of the decoder.

respect to the latent vectors $LS_{\mathcal{T}}$. In practice, we use the latent vectors produced by a single forward pass as initial guess for $LS_{\mathcal{T}}$, and minimize the previous energy function using Adam optimizer.

As we show in Section 5, this step can significantly improve the shape matching results. We remark that this is possible only thanks to the registration formulation of our approach. Shape alignment in 3D space can be optimized continuously, while a point-to-point assignment is non-differentiable due to the combinatorial nature of the problem.

# 5   Experiments & Results

We evaluate the effectiveness of our architecture on a number of challenges. We begin by analyzing the key components of our model, motivating our architectural choices through ablation studies. Finally, we present our results in the context of matching, registration, and inter–class registration.

## 5.1 Experimental settings

**Training data** For all our experiments in the humans domain, we trained our method on the same shapes from the SURREAL dataset [55] used in [34]. It consists of 10000 point clouds for training. Each point cloud has 1000 points, which simulate a significantly sparse sampling of the original shape. During training we augment the data by randomly rotating shapes along the second axis. We also trained our model on the ShapeNet dataset [8]. This dataset is composed of 16881 point clouds representing 3D shapes from 16 different objects categories(from chairs to airplanes), that also do not share a ground truth correspondence. Hence, we trained our method in an unsupervised manner for this particular data. We sample 1024 points for each object and use the same train/test split as in [58].

We also compare our model on other human datasets, thus assessing the ability to generalize to data out of the training distribution. A popular dataset to analyze real identities and poses is FAUST [7], which is composed by ten subjects in ten different poses. We also used a 1000 points version of it, which we refer to as *FAUST1K*. To simulate the noise produced by a 3D acquisition pipeline, we considered the same data from [34] in which the points are perturbated by a Gaussian noise. Also, we challenge our method on *SHREC*'19 [36]. Such dataset is composed by 44 shapes which have different connectivities, poses and densities. In all the experiments we refer to our method as *Our*.

In our comparisons we train our model for 10000 epochs using Adam optimizer [28]. We use 32 latent probes of dimension 64, and 8 layers for both the encoder and the decoder.

**Competitors** We consider 3DCoded [19] (*3DC*) as our principal competitor. Similarly to us, it aims at deforming a shape into another using an autoencoder architecture. However, it assumes one of the two shapes to be a predefined template, limiting its generalization capability. Further, we consider the Linearly-Invariant Embedding approach of [34] (*LinInv*), which learns an high-dimensional embedding in which the shapes can be aligned by a linear transformation. It is based on PointNet, and do not exploit any local structural mechanism. Finally, we considered the Geometric Functional Maps definition [13], using DiffusionNet [53] as feature extractor (*DiffNet*). Similarly to [34], it learns to embed each shape point into a common higher-dimensional.

Both our method and 3DC can improve the registration with a post-processing *refinement*, we refer to these as $3DC_R$ and $Our_R$.

## 5.2 Ablation and analysis

Here we justify our choices on hyperparameters through an ablation study, and we present some insight on the key properties of our method with an in depth analysis.

**Ablation study** To investigate the different components of the proposed architecture, we run a batch of experiments in a reduced version of the SURREAL dataset, using the first 1000 shapes for training, and a different set of 2700 shapes for testing. We also test on FAUST1K.

We report the results in Table 1. We test for three possible ablations: number of latent space vectors, their dimension, and the number of encoder–decoder blocks. We remark that the while SURREAL share discretization and density with the training data, FAUST1K does not. This may explain the difference in performance we observe, and their almost inverse relationship. In fact, it seems that a low number of latent vectors tend to overfit the specific sampling seen during training, while a higher number seems to provide better generalization. A similar behavior can be observed for the other hyperparameters considered. Finally, to reach a good trade off between bias and variance, we choose the configuration that produces the minimum mean error on these two evaluation sets.

**Analysis of the Surface attention** Quantitative and qualitative results showing the importance of the surface attention mechanism compared to the standard point attention. This novel kind of attention we propose has the merit of being much more agnostic to the particular discretization and density of the given point cloud. We can observe this behavior clearly in Figure 3. Here we visualize the attention across different densities and discretization strategies and two different settings, one with surface attention and one with regular attention. With the regular attention mechanism the part of the point cloud attended shows erratic behavior, with different intensities and often even different part that gets attended. Surface attention completely solves this issue, enabling the architecture to achieve

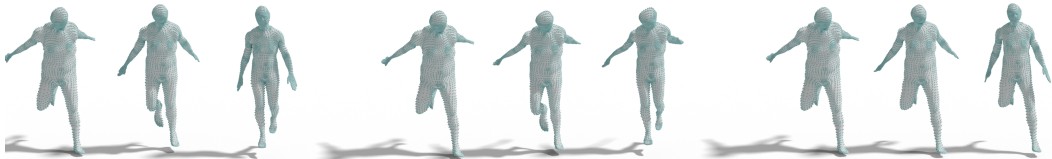

Figure 5: Interpolation examples of two shapes. Left group: left and right-most shapes are the registration of a source shape $S$ to two different target shapes $T_1$, $T_2$, the central shape is obtained passing as input to the decoder latent vectors obtained by linearly interpolating the latent vectors of the two target shapes. In the second and third groups the interpolation is performed only on a subset of the vectors: we fix the latent vectors of the first shape which attention, on the last layer of the decoder, focuses respectively on the upper part of the body (center) and on the legs (right), while interpolating the remaining ones.

greater generalization capacity and enjoying increased robustness, and allowing us to decrease the error on the full version of FAUST of more that $50\%$.

**Multivariate Latent Space**

Inspecting the cross attention of the decoder layer we can seize the impact of the latent probes in our learning. In the top row of Figure 4 we can see how the attention in the first layer of the decoder captures global information of the shape, while in the last layer (bottom row), attention puts its focus on small details of the shape.

The possibility of directly visualizing attention maps and the multivariate nature of the latent space permits further analysis. In Figure 5, in particular, we explore the structure of our latent space and how attention influences it. Similarly to recent non-rigid shape autoencoders [10, 35, 4] we can manipulate the latent vectors to generate new shapes. Specifically, we grab the two latent spaces output of the encoder by registering a source shape $S$ to two different target shapes $\mathcal{T}_1, \mathcal{T}_2$. Starting from the left triad, the left most and the right most shape represents the registration of $S$ to $\mathcal{T}_1$ and $\mathcal{T}_2$ respectively; the one in the middle is a obtained by linearly interpolating the two latent spaces. In the middle triad we add a twist to this procedure, namely we locate the latent vectors attending the most to the upper body of the shape and keep them fixed. A similar procedure is undertaken in the right triad, but this time focusing on the legs. From this experiments, it follows that our latent space is not only linearly navigable, meaning that a linear interpolation of the shapes encoded in the latent space preserve a reasonable semantics, but also, and most interestingly we might say, attention characterizes this space and directly allow for meaningful alteration, or preservation, of selected chosen characteristics.

### 5.3 Results and Applications

In this section we present results and application of our method. In particular, we show state of the art performance on the shape matching task as well as shape registration.

**Matching** One of the task we consider is that of matching. Given two generic point clouds we want to find correspondences between them. Our model approach this task in a natural and elegant way, by registrating one shape onto the other. It becomes trivial then to obtain correspondences through a nearest point search. Results are reported in Table 2. Our method consistently outperforms the state of the art by a solid margin. Furthermore, we notice that our method is endowed with a much greater ability to generalize. This can be noted on the SHREC'19 dataset, as visualzied in Figures 6. The quality of our matching enable us to achieve high quality texture transfer as shown in Figures 1 and 7.

**Template Registration** A classical problem in Computer Graphics is to register a given template, usually a triangular or polygonal mesh, to some acquired point cloud. This setup is a special instance our method, in which the point cloud to be given as input to the decoder remains constant. Regarding our competing methods, even if DiffusionNet and LinInv are not proper registration algorithm, we can move a template point to the corresponding point (as found by the matching algorithm) on the input point cloud. This partially explains why, even though performing the worst overall, achieve lower chamfer distance, since their error is somewhat bound. On the other hand, 3DC is trained

Table 2: Comparison of the average geodesic error on different datasets. FAUST(1k) is obtained from FAUST sampling 1k points, FAUST(1k-noise) is obtained as FAUST(1K) but perturbing each vertex with Gaussian noise. FAUST [7] and SHREC19 [36] have very different sampling densities, with point clouds ranging from $\sim 5$ to $\sim 200$ thousand points.

| Method | FAUST | FAUST(1k) | FAUST(1k-noise) | SHREC19 |
|---|---|---|---|---|
| 3DC | 0.0776 | 0.0542 | 0.0712 | 0.2138 |
| DiffNet | 0.0656 | 0.0534 | 0.0985 | 0.1509 |
| LinInv | 0.0942 | 0.0471 | 0.0618 | 0.1284 |
| **Our** | **0.0513** | **0.0419** | **0.0510** | **0.0802** |
| $3DC_R$ | 0.0485 | 0.0367 | 0.0526 | 0.1935 |
| **Our**$_R$ | **0.0369** | **0.0263** | **0.0410** | **0.0615** |

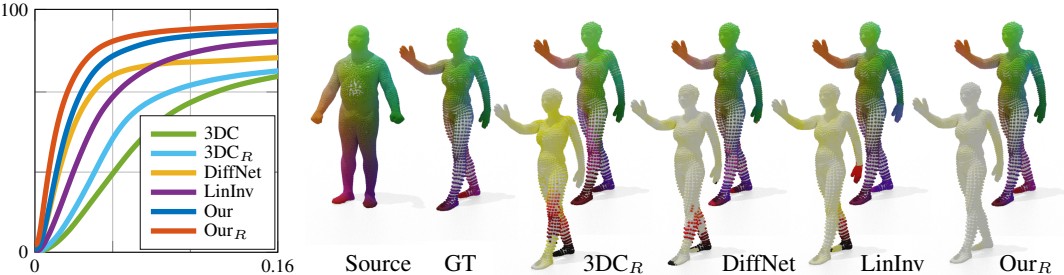

Figure 6: Comparison of different methods on SHREC19 [36]. **Left**: Each curve shows the percentage of points (*y-axis*) with at most a geodesic error (*x-axis*). **Right:** Qualitative comparison. From left to right, the source shape $\mathcal{S}$, the ground truth color transfer to the target geometry $\mathcal{T}$, the results of the competitors and our result. The color transfer predictions are paired with the corresponding error visualizations, from white (error=0) to black (error>0.75).

Table 3: Comparison on the registration task on FAUST [7]. **Left**: Each curve shows the percentage of points (*y-axis*) with at most that geodesic error (*x-axis*). **Right**: Table showing for each method: the mean geodesic error (MGO) of the resulting matching; the Chamfer distance, the maximum and the mean Euclidean distance (Max EU, Mean EU) between the registered template and the target.

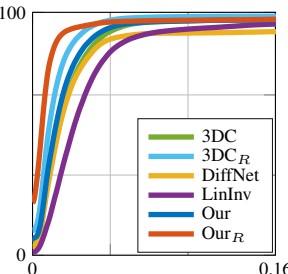

| Method | Chamfer | Max EU | Mean EU | MGO |
|---|---|---|---|---|
| 3DC | 0.0409 | **0.2231** | 0.0723 | 0.0463 |
| DiffNet | **0.0164** | 1.2942 | 0.1023 | 0.0761 |
| LinInv | 0.0177 | 0.3314 | 0.1044 | 0.0692 |
| **Our** | 0.0333 | 0.2299 | **0.0650** | **0.0434** |
| $3DC_R$ | 0.0214 | 0.1705 | 0.0445 | 0.0293 |
| **Our**$_R$ | **0.0129** | **0.1626** | **0.0306** | **0.0275** |

exactly in this fashion. Note also that 3DC is the only method that sees the template shape during the training phase. Even though we train our method on a different task, we manage to improve on the state of the art as can be seen in Table 3, without the need for any, altough possible, fine-tuning.

## 5.4 Unsupervised Registration and Interpolation

One of our main advantages is that we do not require a template. Fixing a common template is not trivial, if not possible at all, when dealing with very different objects. To show the ability of our method of dealing with this challenging scenario we trained a model to register pair of shapes belonging to possibly different object categories of ShapeNet, using the chamfer loss defined in Section 4.

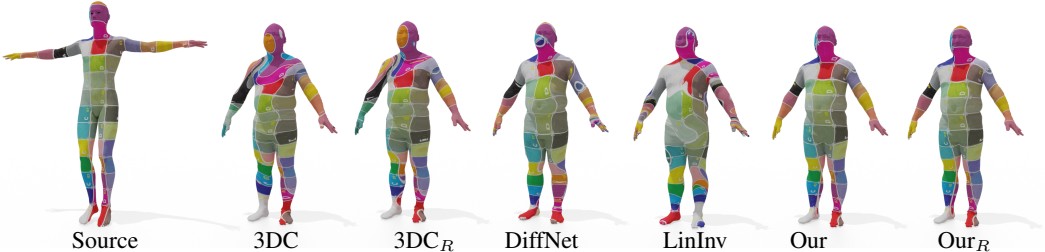

| Source | 3DC | $3DC_R$ | DiffNet | LinInv | Our | $Our_R$ |
|--------|-----|---------|---------|--------|-----|---------|

Figure 7: Qualitative comparison of texture transfer on the SHREC19 [36] dataset. From left to right, the source shape $\mathcal{S}$, the texture transfer to the target shape $\mathcal{T}$ of competitors and our results.

We show in Figure 1 (2nd and 3rd row) two interpolation sequences between two airplanes and between a chair and a table, showing that our method is able to register different objects preserving a meaningful correspondence, represented by similar colors. The interpolated reconstructions are obtained by embedding the outermost shapes in the latent space through the encoder and then using the linearly interpolated latent vectors and the left-most shape as input to the decoder.

# 6 Conclusion

We propose the first transformer based architecture to tackle the problem of non–rigid registration. We introduce a novel surface attention mechanism better suited to exploit the local geometric priors of the underlying structure. Our method reaches state of the art performance in shape matching and shape registration without assuming any fixed template, and generalizes also to different and complex geometries, e.g. handling multiple classes of ShapeNet [8] simultaneously. The attention mechanism at the core of our architecture has the potential to enforce *local control* of the interpolation, as seen in Figure 5.

**Limitations and future works.** Our method shares a common drawback with most transformed-based architectures, requiring long training and post-processing time due to the nature of the refinement procedure. Further investigation is needed to explore the possibility to introduce additional priors on the attention to force a semantically meaningful localization and interpolation behavior.

# 7 Acknowledgment

This work is supported by the ERC Grant No. 802554 (SPECGEO) and the MIUR under grant "Dipartimenti di eccellenza 2018-2022".

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
