# Supplementary Materials
# Shape Registration in the Time of Transformers

**Giovanni Trappolini**
Department of Computer Engineering
Sapienza University of Rome
giovanni.trappolini@uniroma1.it

**Luca Cosmo**
Department of Computer Science
Sapienza University of Rome
luca.cosmo@uniroma1.it

**Luca Moschella**
Department of Computer Science
Sapienza University of Rome
luca.moschella@uniroma1.it

**Riccardo Marin**
Department of Computer Science
Sapienza University of Rome
marin@di.uniroma1.it

**Simone Melzi**
Department of Computer Science
Sapienza University of Rome
simone.melzi@uniroma1.it

**Emanuele Rodolà**
Department of Computer Science
Sapienza University of Rome
emanuele.rodola@uniroma1.it

## 1    Architecture and Implementation Details

In this section, we describe in detail the proposed architecture and its implementation.

Our architecture is composed by an encoder and a decoder.

The encoder receives as input a predefined number of learnable latent probes $LP$, together with the point coordinates of the target point cloud $X_{\mathcal{T}}$. Each layer of the encoder performs an operation of cross-attention between $LP$ and $X_{\mathcal{T}}$ followed by a self-attention on $LP$. Each attention is followed by a feed-forward layer. Before the cross-attention, input 3d-coordinates are embedded in a higher dimensional space through an MLP. We also use positional encoding on $LP$. The output of the encoder is a list of latent vectors $LS_{\mathcal{T}}$ of the same size of the input latent probes.

Specularly, the decoder receives as input the source point cloud 3d coordinates $X_{\mathcal{S}}$ and the latent vectors $LS_{\mathcal{T}}$, and is composed by layers that perform operations of cross-attention between $X_{\mathcal{S}}$ and $LS_{\mathcal{T}}$ and self-attention on $X_{\mathcal{S}}$, each followed by a feed-forward layer. $X_{\mathcal{S}}$ are also embedded in a higher dimensional space by an MLP layer, which, conversely from the encoder, shares weights among all layers. We also use positional encoding on $LS_{\mathcal{T}}$. The output of the decoder goes through a final MLP before outputting the new 3d coordinates of the input pointcloud $X_{\mathcal{S}}$ registered to the target $X_{\mathcal{T}}$.

In our implementation we use 32 latent probes of dimension 64. Point embedders are composed by four linear layers of size $(8, 16, 32, 64)$ interleaved by ReLU activation. As standard in transformers, the feed–forward layer is made of two linear layers of size $(512, 64)$ interleaved by ReLUs. The final MLP block is composed of five linear layers of decreasing sizes: $(48, 24, 12, 6, 3)$ also interleaved by ReLUs. We also use multi-head attention with 4 heads. The encoder and decoder blocks, light grey in Figure 2 in the main paper, are repeated 8 times each, weights are not shared.

When performing matching, we switch the target and source shape and we pick the version minimizing the registration's chamfer distance error.

35th Conference on Neural Information Processing Systems (NeurIPS 2021), Sydney, Australia.

The model was trained using an NVIDIA 2080ti. Keops [3] was used to improve the scalability capacity of the model. In fact, thanks to its peculiar memory efficient implementation, it enables the processing of shapes with more than 200 thousand points (SHREC'19 dataset).

## 2 Refinement Procedure

As described in the Methods section in the main paper, the peculiar structure of our architecture allows us to refine the registration results. Here we provide additional details on this procedure.

A key choice in the refinement process is the number of refinement steps. As can be seen in Figure 1, the graph suggests there is a large decline in geodesic error during the first twenty iterations, while lower marginal returns are obtained after this threshold. This is confirmed visually with the first refined registration (25 steps) having the largest decrease in error. Later iterations however contribute to more visually appealing registrations. In all experiments we perform 100 refinement steps for both the refined versions of 3DC and our method. The optimization is performed using the Adam optimizer with a learning rate of $5e - 3$.

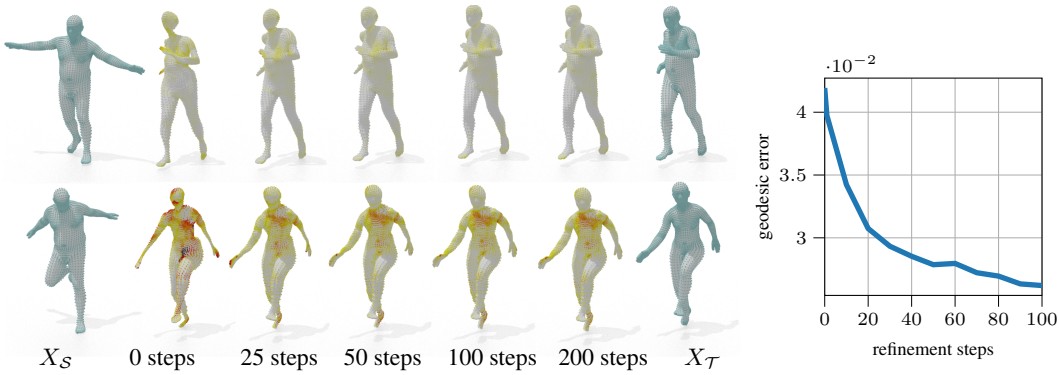

$X_{\mathcal{S}}$    0 steps    25 steps    50 steps    100 steps    200 steps    $X_{\mathcal{T}}$

Figure 1: On the left, a pair of qualitative examples showing the registration quality improvement using an increasing number of refinement steps of our algorithm. On the right, the mean geodesic error of our method with different number of refinement steps on FAUST1K.

## 3 SMAL Dataset

We report qualitative results on animal shapes, confirming the ability to generalize to other types of shapes other than humans and rigid objects.

The SMAL [6] model provides the equivalent of SMPL for several animals. We employed this model to generate 20300 shapes of different animals. We trained on 20000 examples and tested on the remaining 300 datapoints.

In Figure 2, we can observe the results we obtain with our method on this dataset. On the extremities we have the source (left) and the target (right) shapes, colored accordingly to the predicted matching between the two. The shapes in the middle are gradual interpolations of the two, obtained by linearly interpolating their latent representation. Thanks to the great flexibility of our architecture, we can generate smooth transitions and obtain high quality matching between animals of different classes in different poses.

## 4 Outliers

We have tested our model resilience to outliers using the same setup and data proposed in LinInv [4], consisting in a strong Gaussian perturbation (standard deviation of 0.03) applied to the point positions of the input point clouds.

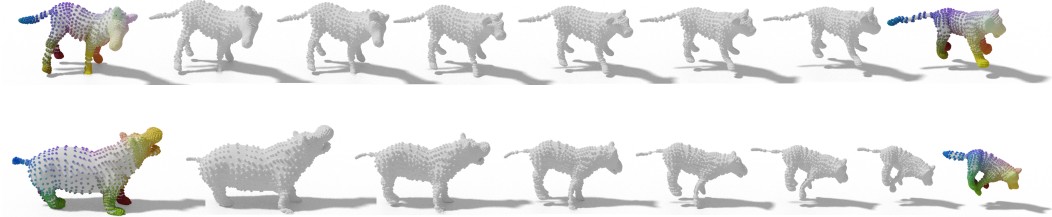

Figure 2: Interpolation examples on two pairs of animals belonging to different classes: horse-lion (first row), hippo-dog (second row). These results are possible only thanks to the high-quality registration provided by our method.

We report in Table 1 quantitative performance for all the competitors in this setup, showing that our method is resilient to noise and manages to outperform all other methods. We also show some qualitative results in Figure 4 (last row) and in Figure 7 (last two rows).

Table 1: Comparison of the average geodesic error on FAUST(1k-outliers). This dataset is obtained sampling 1k points from FAUST and perturbing each vertex with Gaussian noise with high standard deviation (0.03).

| Method | FAUST(1k-outliers) |
| --- | --- |
| 3DC | 0.2306 |
| DiffNet | 0.3509 |
| LinInv | 0.1738 |
| **Our** | **0.1657** |
| $3DC_R$ | 0.2101 |
| **Our$_R$** | **0.1479** |

## 5   Additional Results and Visualizations

In this section, we report further qualitative results and comparisons with other methods.

**Out of distribution samples**   In Figure 3 we show results involving shapes coming from different datasets, being significantly different from the ones observed during training. In particular, we show a registration from TOSCA's alien and a FAUST shape, and vice-versa. We see that our method provides a good correspondence in both cases. Then, we also tested on two statues from the Scan the World project [1], these two present clutters and topological noises (e.g. the glued hands of the first or the beard of the second). In the second row of the same image, we report an experiment with partiality (the backside of the shape is absent). Notice that partial shapes were not seen at training time.

**Registration**   In Figure 4, 5 and 6 we show some registrations results. Notice that DiffNet and LinInv output point-to-point correspondences which naturally lie on the surface of the target shape, while registration methods have to find the correct alignment. Our registration provides a better alignment of the shapes and more accurate reconstructions of the finer details (e.g. the shape of the heads in the first row or the torso in the second one).

**Matching**   In Figure 7 we report some shape-to-shape matching on FAUST 1K (in the first two rows) and on the outliers dataset proposed in [4] (in the last two rows). For each pair, we visualize the matching as color transfer, showing from left to right, the ground-truth, the outcome of 3DC without and with refinement, DiffusionNet, LinInv, and our method without and with refinement. For our method, we observe a general resilience to sparsity and noise.

In Figure 8 we collect further color transferring qualitative results, with also the matching error reported as hot map on the point clouds. In Figure 9, we visualize some results in the same setup but with additional noise on the point clouds.

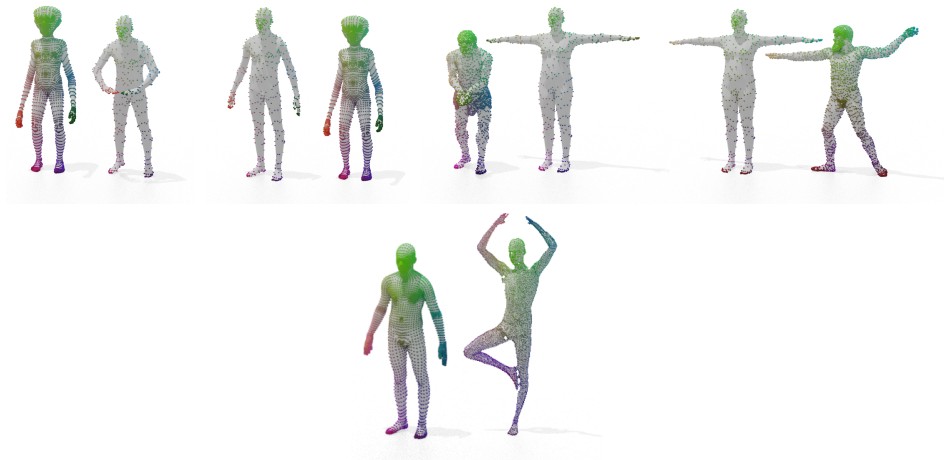

Figure 3: Qualitative results on shapes that present significant differences to the training examples. We report registrations between a FAUST shape and an alien from TOSCA, two statues from the Scan the World project, and a partial shape (second row).

Finally, in Figure 10 we report a texture transfer performed between two shapes from SHREC19 dataset [5].

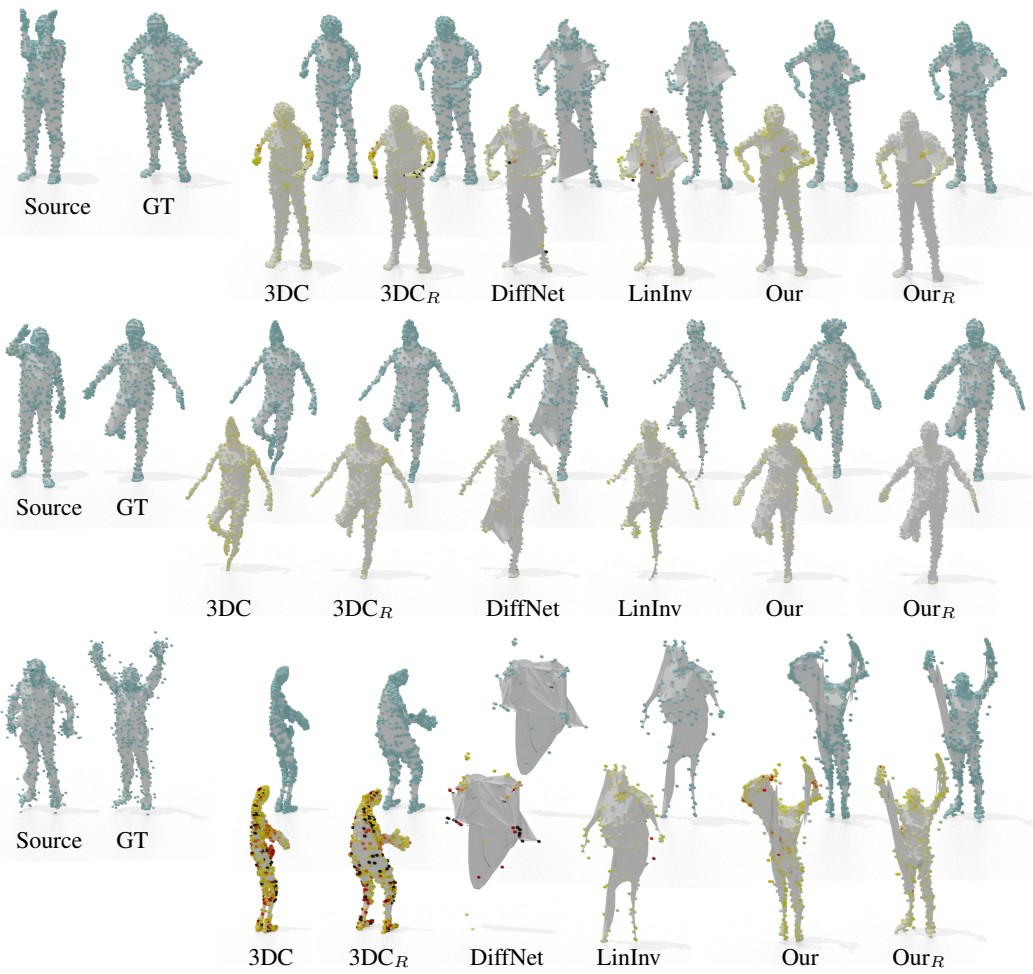

Figure 4: Qualitative comparisons of arbitrary shape registration on the FAUST (1k) [2] dataset *(first two rows)* and the dataset of outliers from LinInv [4] *(last row)*. The registration are performed between two arbitrary shapes, however, 3DC can only register the template and not the source shape into the target shape.

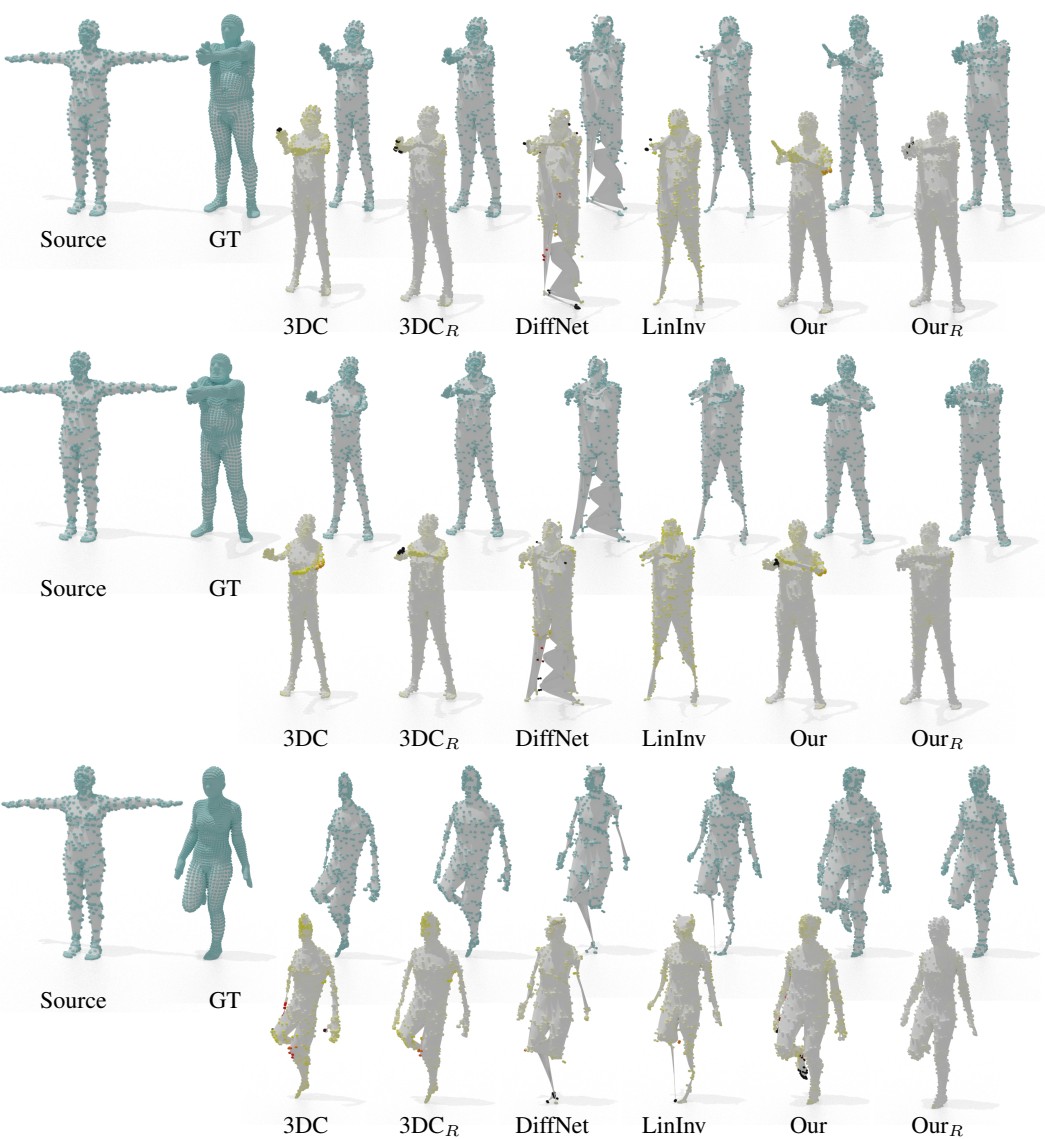

Figure 5: Qualitative comparisons of template registration on FAUST.

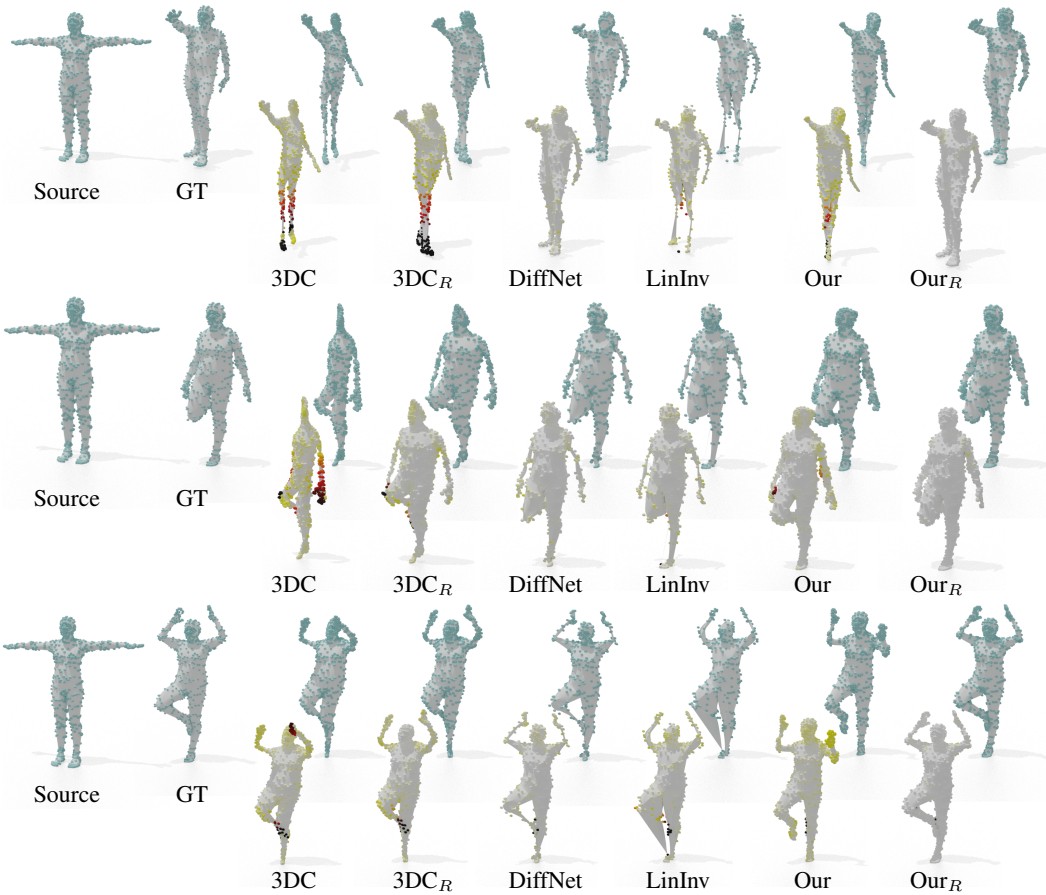

Figure 6: Qualitative comparisons of template registration on FAUST (1k).

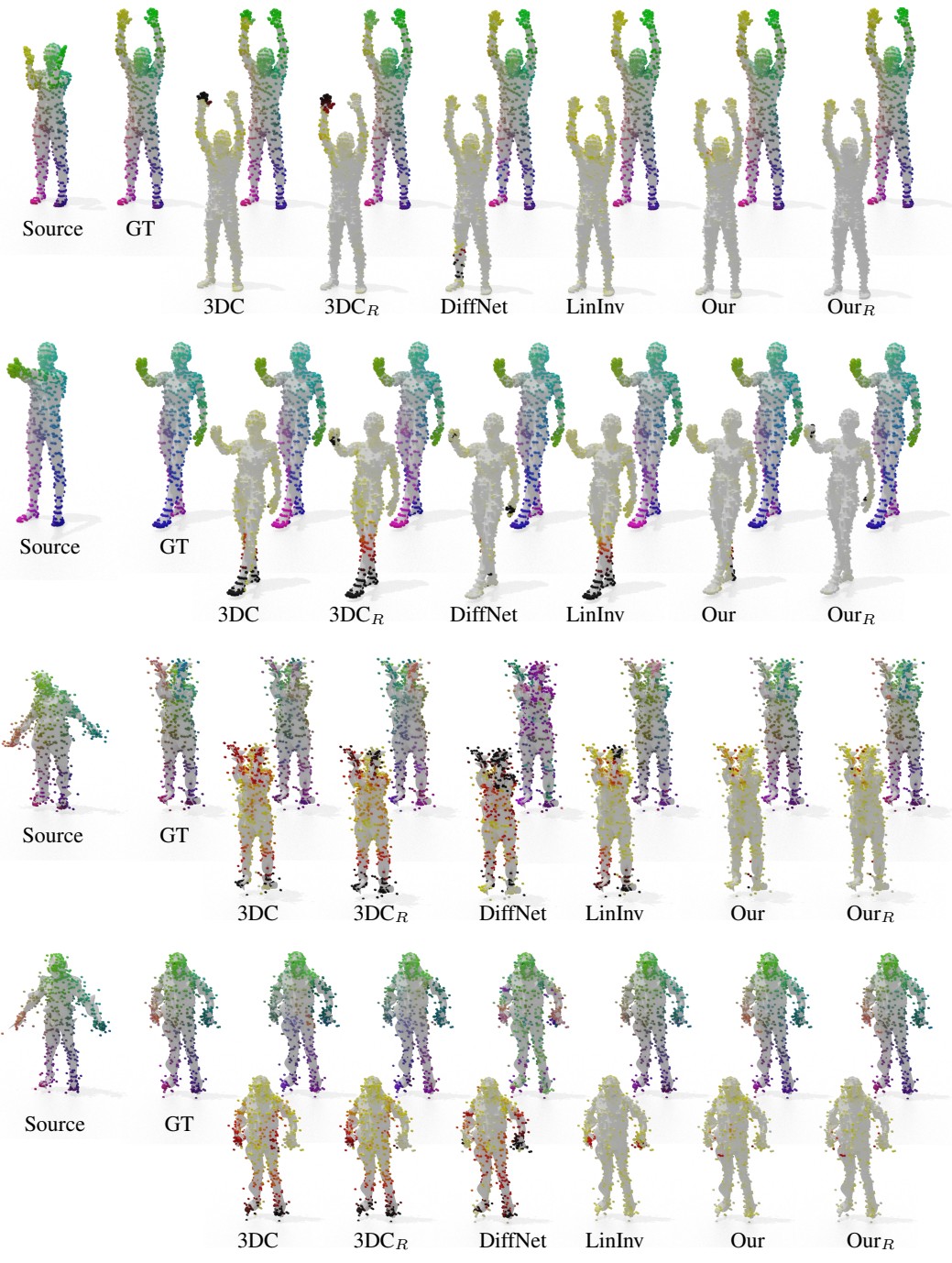

Figure 7: Qualitative comparisons of shape matching on the FAUST (1k) [2] dataset *(first two rows)* and the dataset of outliers from LinInv [4] *(last two rows)*.

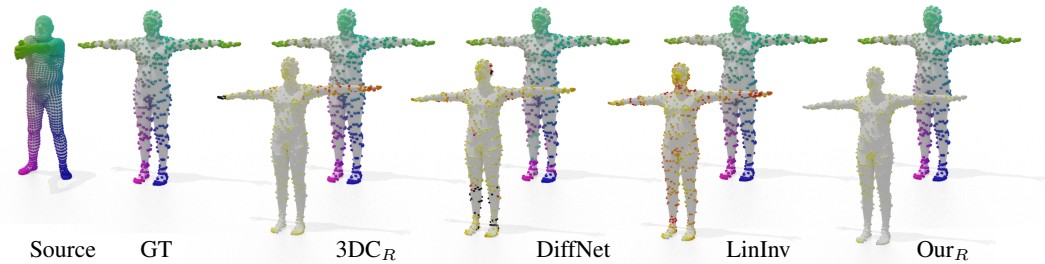

Figure 8: A qualitative comparison of template registration on FAUST [2]. In particular, *3DC_R* exhibits an error due to topological changes between $\mathcal{S}$ and $\mathcal{T}$, *LinInv* shows small but widespread errors.

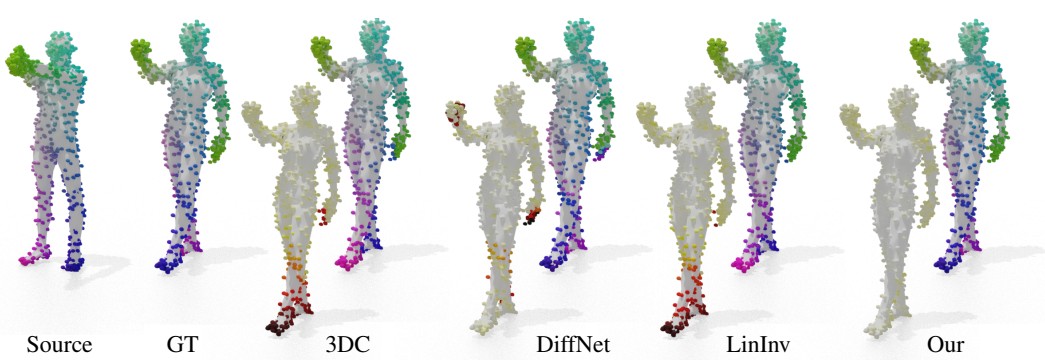

Figure 9: A qualitative comparison of shape matching on the perturbed FAUST [2] dataset.

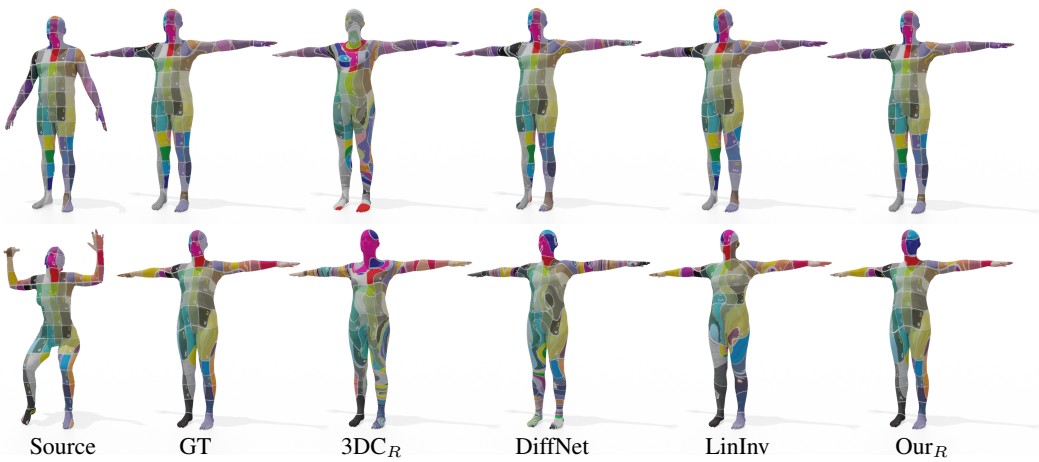

Figure 10: Two qualitative comparisons of texture transfer on the SHREC19 [5] dataset. From left to right, the source shape $\mathcal{S}$, the ground truth transfer to the target geometry $\mathcal{T}$, the results of the competitors and our result. In the top row we report an *easy* example of texture transfer, where almost all competitors perform reasonably well. In the bottom row, we report an *hard* example of texture transfer caused by the large variation in the pose, the missing parts (fingers) and the presence of extra object (in the hand of the source shape).