# OpenReview forum: "Shape Registration in the Time of Transformers"
_NeurIPS.cc/2021/Conference — NeurIPS 2021 Poster_

### Official Review · Reviewer_NiyT · 2021-07-11

**Rating:** 6
**Confidence:** 4

**Summary:**

Summary:

a) This paper proposes a transformer-based procedure to register non-rigid 3D point clouds or shapes;
b) Given a source point cloud, this paper aims to deform its coordinates to align to target point cloud;
c) Authors modifies the attention mechanism to make it aware of the point cloud density, by adding weight to the original attention score;
d) Experiments show the advantages of this paper over previous methods.



**Limitations And Societal Impact:**

Yes.

**Main Review:**


i) Strong points.

a) The use of the Transformer architecture for the task of non-rigid registration is new,  and seems to be the first as claimed by the authors;
b) For complete shape scans, it makes sense to modify the attention mechanism to make it aware of the point cloud local density;
c) Overall, this paper is well-written, and easy to follow.

ii) Weak points.

a) Architecture.

As noted by authors in line 109, the deep closest point [50] also use a transformer to register rigid shapes. In light of this,  it is highly beneficial if the author can provide the  following comparisons:

  * Trim the architecture of [50], add MLP to \Phi_X or \Phi_Y (Figure 2 in [50]) to estimate the deformed points.

  * Use the above baseline to compare with your proposed architecture.

b) Partial scans.

This paper lacks experiments of partial--to-partial registration of non-rigid point clouds. Adding noise in experiment is good (Line 207).  I would expect this partial-to-partial registration are more  realistic in many real-world applications.   How will your method handle  significant portion of non-matchable outliers?

Furthermore, the above experiment would also check the effectiveness and robustness of the proposed modified attention mechanism (Equation 1).

c) Quantitative comparisons between original attention and modified attention.

Giving qualitative comparisons in Figure 3 is good. However, I would like to see full quantitative comparisons besides the 50% claim on line 245.

d) Computation time.

Please specify the details of computation times, and compare them with respect to other methods (in Table 2).
As indicated on Line 301-302, the proposed method is computationally expensive/time consuming.  Why do you claim it is an "efficient registration" method (c.f. Line 1) ?




**Time Spent Reviewing:**

4 hours

---

> ### Author Response · Authors · 2021-08-10
> **Response to reviewer NiyT**
>
> We would like to thank the reviewer for the insightful comments, in particular for its remarks on partial scans.
>
> We would like to clarify some concerns:
>
> 1. **Architecture.** As noticed by the reviewer we did not originally consider [50] as a comparison since their method deals with rigid transformations. To the best of our understanding, the proposed formulation would consist of an embedder (either PointNet or DGCNN) followed by a transformer decoder layer and an MLP. This would be semantically, and in purpose, very similar to a standard transformer. As answered to reviewer VAnV (see point 1), we believe that embedding the shape in a latent space is a better and more versatile design choice.
> 2. **Partial Scans.** We thank the reviewer for raising this interesting point of discussion. While we have not performed partial to partial registration of non-rigid point clouds, we did perform a full to partial registration of this sort in the second row of figure 10 of the supplementary material. The results are qualitatively good, even the more considering that no partial shapes are observed at training time. Unfortunately, our method does not work directly in the partial-to-partial setting, and it would probably need to adopt some specific design choices. We believe this is indeed an interesting future research direction.
> 3. **Surface vs Classical attention.** On the full FAUST dataset, the unmodified attention mechanism averages a 0.0781 geodesic error, while the modified surface attention mechanism averages a 0.0513 geodesic error. If needed, we will run and report results without surface attention for all datasets in the final manuscript.
> 4. **Computation time.** While our method is computationally expensive and time-consuming during the training phase as outlined in section 5.1, at test time the registration between two shapes can be achieved through a simple forward pass of the model, at least in our non-refined version. This is in contrast to classical methods in the literature that require long inferential time, and in line with competitors (not considering the slower [46]) which however show worse performance. Such trade-off (inefficiency at training time, efficiency at test time) is the preferable one to enable many applications since it saves resources at inference time.

---

### Official Review · Reviewer_eCbC · 2021-07-15

**Rating:** 7
**Confidence:** 4

**Summary:**

The authors propose a novel transformer based approach for 3D shape registration for non-rigid objects.
Transformers have been very successful in language modelling and recently also in vision applications. This paper introduces transformers for 3D shape registration which is an interesting step forward.
The approach also outperforms competing baselines.

**Limitations And Societal Impact:**

Authors very briefly mention some limitations of their work but this can be further improved (see comments above).

**Main Review:**

+ The approach is technically sound and makes intuitive sense.
+ Quantitative results look good.
+ Authors will release their code and data, which is appreciated.

I have minor questions regarding the work:

1. Surface attention, Eq. 1. The whole buzz around the transformer idea is centered around the fact that we no longer need to provide inductive bias to the network regarding how different components in the input are related. Eg: with CNNs the inductive bias takes the form of predefined kernel shapes that indicate to the network that some correlation exists within the receptive field of the kernel. Whereas with transformers no such assumption needs to be made. Despite this fact, why does the proposed method need explicit normalization for point density? Since SURREAL is a synthetic dataset, why can't it be augmented such that network learns to generalise to different sampling strategies. This is interesting because despite the proposed normalisation, there is a performance gap between performance on SURREAL and FAUST, which authors attribute to different surface distributions.

2. Authors mention that their method can be used to obtain correspondences between two shapes. Why is this not evaluated on the FAUST challenge?

3. Table 2: Are the baselines trained on the same data as the proposed method?

4. Authors currently restrict their problem setting to low resolution pointclouds (1000 points as input). How do the transformers perform with denser inputs?

5. Current limitations stated in the conclusion section are generic. Since this is a novel direction, it will be helpful to add a dedicated limitation and future works section with more detail. It would be good to also discuss other directions in 3D vision where transformer architectures would be beneficial.

Overall, I like the idea of using transformers for 3D shape registration and the authors also show good quantitative results.


**Time Spent Reviewing:**

3

---

> ### Author Response · Authors · 2021-08-10
> **Response to reviewer eCbC**
>
> We would like to thank the reviewer for the insightful comments, in particular for pointing out possible future directions in 3D vision.
>
> We would like to clarify some concerns:
>
> 1. **Surface Attention.** While transformers in other subfields do take pride in not needing inductive biases, we believe that for the problem of shape registrations this new formulation provides great advantages at no real cost. Surface attention is an inductive bias injected into our model. While, in theory, one could learn this bias from the data, such a procedure would require the model to be exposed to every possible discretisation strategy and density.  Instead, we propose to adopt the weighted attention mechanism. Including the areas makes the attention mechanism theoretically closer to the integral formulation of the inner product defined on surfaces. This formulation manages to greatly improve the generalisation capacity and performance of the model as shown by quantitative and qualitative results. We emphasise that this is only possible thanks to the flexibility offered by the transformer architecture. At the same time, we partially overcome one of the crucial drawbacks of transformers - i.e. training time -  reducing the quantity of data required for training.
> 2. **Faust challenge.** While indeed the FAUST challenge has been an interesting use case for several years, we believe that it has been saturated and it is standard for recent correspondence methods not to test themselves on that dataset [30, 13]. On the other hand, we consider significantly more interesting the SHREC dataset, which addresses a significantly wider set of challenges (and also includes some shapes from the Faust dataset).
> 3. **Baselines**. All baselines are trained on the same data and under the same augmentations as the proposed method, we’ll make it clear in the text.
> 4. **Low Resolution.** The self-attention mechanism in the decoder scales quadratically with the cardinality of the point cloud. Even if we can scale much further than 1000 points, this becomes less necessary thanks to the generalisation capacity of our model induced by the surface attention. Moreover, at testing time we can process arbitrary point cloud dimensions, as in SHREC where some shapes contain around 200k points.
> 5. **Limitations and future directions.** As pointed out in other responses, while we do mention the limitations of our method we’ll make sure those are more evident and appropriately discussed. As for future research directions, we believe the three main focus points are: a) Architecture: even though we based our model on a state of the art transformer architecture, we believe that innovative modules or more sophisticated feature extractors could be better suited to this domain and could further improve performance. b) Attention mechanism: our new formulation shows that the underlying geometry is to be kept into consideration. There are many ways in which this could be expanded to fit the 3D domain; i.e. mesh information, the use of differential operators. c) Multimodality: transformers offer a great level of flexibility. This could be exploited to formulate new ways of matching different objects; i.e. a 3D object and an image.

---

### Official Review · Reviewer_VAnV · 2021-07-16

**Rating:** 7
**Confidence:** 4

**Summary:**

This paper proposes a Transformer-based point cloud registration scheme. Given two point clouds, the target and an initial latent feature go through the encoder of a Transformer-like network, and the resulting latent vector is combined with the source and goes through the decoder to yield a transformed point cloud (that resembles the target). In the Transformer-like structure, iterative cross attention is used, which is inspired by [21]. There are some other details to handle the particular problem efficiently, e.g., density aware attention for handling the density differences in point clouds. The decoder and the loss function can also be used to refine the result with further optimization. The experiments show that the proposed method mostly achieves state-of-the-art results (sometimes with large gaps) for well-known datasets.

**Limitations And Societal Impact:**

** What happens if we use somewhat heterogeneous point clouds for source and target? In my understanding, most of the experiments are about more-or-less homogeneous data. Discussing this for exploring the limitations of the method can be interesting.

**Main Review:**

+- Somewhat derivative, but appropriately designed: The main ideas of the paper are inspired by other works, such as the recently popular Transformer structures and especially the iterative cross attention structure [21]. Architecture-wise, one might say that the work is somewhat derivative from [21]. But at the same time, some additional effort (e.g., density-adaptive attention) is appropriately made to handle the problem effectively. The method yields state-of-the-art performance, and sometimes, the gaps are quite large, which suggests that the proposed method is indeed effective.

** Why the Transformer for cross-modality?: The iterative cross attention inspired by [21] is originally for cross-modality. Point clouds are rather homogeneous in this sense. In my understanding, the authors are using this to easily pass on the latent feature (which can be smaller and contain essential information) from the target to the source. This can be indeed an efficient way to handle the problem. But at the same time, I think that a regular Transformer-like structure could also be used and there is a chance that the performance can be better. Why is the proposed method a better choice? More discussion is needed.

** How is the initial latent feature (latent probe) handled?: One thing that is not clearly explained in the paper is how the latent probe, which is one of the inputs of the network, is handled. The authors have provided a code as supplementary material, but providing this in the paper or in the supplementary document will be good for completeness.

[After rebuttal]
I have read all the other reviews and the authors' responses. All my previous questions have been answered. I stay with my original score.

**Time Spent Reviewing:**

3

---

> ### Author Response · Authors · 2021-08-10
> **Response to reviewer VAnV**
>
> We would like to thank the reviewer for the insightful comments, in particular for pointing out our use of the latent space.
>
> We would like to clarify some concerns:
>
> 1. **Design choice.** We thank the reviewer for raising this interesting point of discussion. We believe our peculiar formulation grants us considerable advantages. As noted by the reviewer, the latent space can contain information in a condensed form with a dimension independent from the number of input points, producing a compact latent representation of the input point cloud. In fact, as shown in figure 5, we can easily manipulate the latent space in a semantically significant manner (i.e. localized interpolation between two shapes). Furthermore, we can adopt a refinement step on the latent space that greatly improves the model performance. With a standard transformer architecture, we would lose this interesting property since we would not have a compact representation of the shape.
> Finally, this formulation reduces the computational complexity of the model, since there is only one layer that scales quadratically with the number of points in the point cloud, that is the self-attention in the decoder, while under the standard transformer formulation this happens in every attention layer.
>
> 2. **Latent Probes.** The latent probes are part of the network parameters and are learned during the training phase. We will clarify this in the text.

---

### Official Review · Reviewer_pM1i · 2021-07-23

**Rating:** 6
**Confidence:** 1

**Summary:**

This paper introduces the use of transformers architecture for shape matching and the method is applied to 3D point clouds. The main innovation with respect to standard attention architecture on point clouds is the weight introduced in the attention mechanism which accounts for the local density of the shape. Experiments show a clear gain of the method compared to other state of the art method, both quantitatively and qualitatively.

**Limitations And Societal Impact:**

Limitations are not discussed enough, in particular the training time of transformers and ways to get around it.

**Main Review:**

On the positive side, the proposed method seems to outperform state of the art methods in the field, although I am not an expert in this field.
The method does not require the use of a template which makes the method directly applicable.

My main concern is the writing style of the paper which is too loose and not accurate enough in my opinion. The first two sections are nicely written overviews discussing common generalities on attention and shape registration. Beginning of section 4 loosely describes the architecture with inaccuracies in formula (1). It is not possible to reimplement the method from reading out the main text.
The authors do not discuss the problem of long training time of transformers in their situation.

Questions:
- Is the surface attention a particular case of classic attention in a particular formulation? Is it an inductive bias that could be learnt from enough data?
- How is the bottleneck of attention (quadratic in the input) dealt with in the context of hundreds of thousand of points ? it is not detailed enough.

Comments:
- it seems that there are multiple typos in formula 1. The index summation $t$ is not present elsewhere in the formula, and the notations $\exp^{(x)}y$ and $\exp^{xy}$ are quite unusual. I would suggest to replace it with more standard notations.
- The running times and computational resources are difficult to find in the main text although the authors replied positively that it is stated in section 5. I have hard time to find it. It is not discussed in this section and only touched upon in the conclusion. It needs more discussion.
- Chamfer loss is said to be defined in section 4, but it is a pointer to another publication.


**Time Spent Reviewing:**

2.5

---

> ### Author Response · Authors · 2021-08-10
> **Response to reviewer pM1i**
>
> We would like to thank the reviewer for the insightful comments, in particular for the interesting consideration on the surface attention mechanism.
>
> We would like to clarify the following concerns:
>
>
> 1. **Lack of details in section 4.** We included Implementation details in Section A of the supplementary materials. Nonetheless, we will be happy to enrich section 4 with further details exploiting the additional page allowed after rebuttal. We also plan to release to the public the code shared in the supplementary material.
> Furthermore, we thank the reviewer for pointing out the error in equation 1, we will correct it and adopt a more standard notation as follows:
> $$
> \widetilde{W}\_{i,j} = \frac{e^{W_\{i,j}} A(X)\_j} {\sum_t e^{W\_{i,t}}  A(X)\_t}
> $$
> We will also explicit the formula for the chamfer distance.
>
> 2. **Training time and computation resources.** While we state our training time in Section 5.1, we do not actively discuss the problems related to using a transformer architecture, such as the lengthy training time. We will make sure this is appropriately covered in the final manuscript.
> Computational resources are listed in Section A of the supplementary materials. We will include and make them more evident to the reader in the main manuscript.
>
> 3. **Surface attention.** Surface attention is an inductive bias injected into our model. While, in theory, one could learn this bias from the data, such a procedure would require the model to be exposed to every possible discretisation strategy and density.  Instead, we propose to adopt the surface attention mechanism. Including the areas makes the attention mechanism theoretically closer to the integral formulation of the inner product defined on surfaces. This formulation manages to greatly improve the generalisation capacity and performance of the model as shown by quantitative and qualitative results. We emphasise that this is possible thanks to the flexibility offered by the transformer architecture. At the same time, we partially overcome one of the crucial drawbacks of transformers - i.e. training time -  reducing the quantity of data required for training.
>
> 4. **Quadratic Bottleneck.** Thanks to our peculiar formulation, we incur in this bottleneck only in the self-attention layer of the decoder. As explained above, we do not need to train on large point clouds, and at test time we overcome this obstacle using Keops, which allows us to deal with large point clouds (hundreds of thousands of points).

---

### Decision · Program_Chairs · 2021-09-27

**Decision:**

Accept (Poster)

**Comment:**

All reviewers agree that the idea of using a transformer-type architecture for (non-rigid) 3D point cloud registration is interesting and novel. While reviewers have raised questions or pointed out concerns about certain issues (e.g., being derivative), I am of the opinion that the authors sufficiently addressed those points in their answers (despite little or no serious discussion happening). I do, however, encourage the authors to follow the reviewer's advice on improving the writing style in some places and including all required clarifications as promised in the rebuttal.